# Iron Deficiency and Iron Deficiency Anemia: Potential Risk Factors in Bone Loss

**DOI:** 10.3390/ijms24086891

**Published:** 2023-04-07

**Authors:** Jiancheng Yang, Qingmei Li, Yan Feng, Yuhong Zeng

**Affiliations:** Department of Osteoporosis, Honghui Hospital, Xi’an Jiaotong University, Xi’an 710054, Chinaxahhzyh@163.com (Y.Z.)

**Keywords:** iron deficiency, anemia, iron metabolism, osteoblast, osteoclast, osteoporosis, bone loss

## Abstract

Iron is one of the essential mineral elements for the human body and this nutrient deficiency is a worldwide public health problem. Iron is essential in oxygen transport, participates in many enzyme systems in the body, and is an important trace element in maintaining basic cellular life activities. Iron also plays an important role in collagen synthesis and vitamin D metabolism. Therefore, decrease in intracellular iron can lead to disturbance in the activity and function of osteoblasts and osteoclasts, resulting in imbalance in bone homeostasis and ultimately bone loss. Indeed, iron deficiency, with or without anemia, leads to osteopenia or osteoporosis, which has been revealed by numerous clinical observations and animal studies. This review presents current knowledge on iron metabolism under iron deficiency states and the diagnosis and prevention of iron deficiency and iron deficiency anemia (IDA). With emphasis, studies related to iron deficiency and bone loss are discussed, and the potential mechanisms of iron deficiency leading to bone loss are analyzed. Finally, several measures to promote complete recovery and prevention of iron deficiency are listed to improve quality of life, including bone health.

## 1. Introduction

Iron is one of the essential mineral elements for the human body. Since hemoglobin (Hb) synthesis consumes the most iron in the human body to produce 200 billion red blood cells daily [1], anemia is a more obvious sign of iron deficiency, and iron deficiency anemia (IDA) is usually considered to be a synonym of iron deficiency. However, iron deficiency is a broader condition that often precedes the onset of anemia or persists without progression [2]. IDA is a more serious condition of iron deficiency, in which low levels of iron are related to anemia and the presence of microcytic hypochromic red cells [3]. Iron deficiency mainly includes absolute and functional iron deficiency [4]. When the whole body’s iron reserves are insufficient or exhausted, absolute iron deficiency will occur. Functional iron deficiency is a disease in which iron storage of the whole body is normal or even increased, but iron availability for incorporation into erythroid precursors is insufficient, such as thalassemia and sickle cell anemia [5,6]. In this article, we focus mainly on absolute iron deficiency.

Iron deficiency or IDA can affect many organ functions of the human body. For example, IDA is related to functional impairment that affects cognitive development [7], immune mechanism [8], and physical work capacity [9]. The relationship between iron deficiency and bone health comes from clinical observations in IDA patients who suffered bone loss [10,11]. In addition, low iron in diet is also associated with bone loss [12]. In this review, relevant studies on iron deficiency and bone loss are discussed, and the potential mechanisms of iron deficiency leading to bone loss are analyzed.

## 2. Iron Deficiency and Iron Deficiency Anemia

### 2.1. Epidemiology

Anemia is a serious global public health problem, especially affecting young children and pregnant women. Worldwide, it is estimated that 39.8% of children (aged 6–59 months) [13], 29.9% of women of reproductive age (aged 15–49) [14], and 36.5% of pregnant women (aged 15–49) [15] were anemic by the WHO in 2019. According to a estimation by the WHO, 50% of anemia cases in women and 42% of anemia cases in children could be eliminated by iron supplementation [16]. However, a meta-analysis of population studies shows that the contribution of iron deficiency to anemia may be less than the estimate of WHO: 25% for children and 37% for women [17]. This may be because there are few population studies to measure iron biomarkers (except hemoglobin), and the interpretation of iron biomarkers during inflammation is also very complex, so the prevalence of iron deficiency in low- and middle-income countries is uncertain. The prevalence of iron deficiency in children aged 6 months to 5 years was estimated to be 26.1% in Liberia, 20.2% in Cameroon, 18.4% in Laos, 10.6% in Colombia, and 14.8% in Mexico; the adjusted prevalence of iron deficiency in premenopausal women who were not pregnant was 19.9%, 13.7%, 24.0%, 24.1%, and 30.4%, respectively, in the same countries [18]. According to the national health and nutrition examination survey, 11% of children in the USA between the ages of 6 months and 5 years, 15% of premenopausal women, and 18% of pregnant women have iron deficiency [18,19]. Iron deficiency and anemia are more prevalent in underprivileged subpopulations, such as low-income individuals, members of indigenous groups, and refugees and migrants from low-income and middle-income countries [20,21].

### 2.2. Pathophysiology

Adult men contain 35 to 45 mg of iron per kilogram of body weight in a physiologically normal state, whereas premenopausal women have lower iron stores because of regular blood loss from menstruation [22]. More than two-thirds of body iron is contained in hemoglobin of erythrocytes and erythroblasts in bone marrow, circulation, and reticuloendothelial macrophages. Most of the remaining iron is stored in the liver as ferritin, and the rest is found in myoglobin and enzymes [22]. Plasma contains only 0.1% of total body iron, which is bound to transferrin. In this form, iron can be transferred to various tissues via binding to the transferrin receptor. Since iron is a highly reactive metal and can catalyze the formation of toxic free radicals, plasma iron is rapidly turned over. Plasma iron is predominantly derived from iron scavenged by macrophages from senescent erythrocytes, and a small amount (1–2 mg per day) is absorbed from the diet at the duodenum [23]. Due to menstrual bleeding, sweating, skin peeling and urine excretion, about 1–2 mg of iron is lost every day. Since the body lacks a regulatory pathway for iron excretion, a balanced regulation of dietary iron intake, intestinal iron absorption and iron cycling is necessary.

Hepcidin (encoded by *HAMP*), a small peptide derived from hepatocytes, is the central regulator of systemic iron homoeostasis [24]. Hepcidin binds to ferroportin (FPN), which is the only known iron-exporting protein, and is then internalized and degraded by lysosomes. Subsequently, duodenal enterocytes, macrophages, as well as hepatocytes can no longer export iron, which is sequestrated in these cells. Hepcidin expression is downregulated during absolute iron deficiency or periods of increased iron demand, whereas it is upregulated during inflammation and periods of high iron concentrations in the liver and plasma.

Iron deficiency enhances erythropoiesis by increasing renal erythropoietin (EPO) production and erythroblastic EPO sensitivity via the genetic loss of the EPO receptor (EPOR) partner transferrin receptor 2 (TfR2) [25]. In absolute iron deficiency, the reduction of hepcidin production induced the increase in FPN expression of erythroblasts and erythrocytes [26], which led to the release of iron into serum via FPN to alleviate serum iron depletion and protect erythrocytes from oxidative stress [27]. These data show that erythroid cells donate iron to maintain iron supply elsewhere in iron deficiency, thus reducing intracellular iron availability of erythroblasts. These results also explain why IDA is the most obvious early manifestation of iron deficiency in mammals. Due to the increased erythropoiesis and decreased iron supply, heme content per cell is reduced, resulting in an increase in erythrocytes characterized by hypochromia and microcytosis, which eventually leads to anemia (Figure 1). In addition, erythroferrone (ERFE), a hormone produced by erythroblasts in response to EPO, can mediate hepcidin suppression during stress erythropoiesis [28].

### 2.3. Diagnosis of Iron Deficiency and Iron Deficiency Anemia

According to the severity, iron deficiency can be divided into three stages, namely iron depletion, iron deficiency without anemia, and iron deficiency anemia [2,29]. The terminology and cut-off values of biochemical markers for identifying iron deficiency at each stage are shown in Table 1 [2,3,29,30]. In the stage of iron depletion, the body’s iron stores are exhausted, which can be detected by the reduction of serum ferritin. The stage of iron deficiency without anemia is characterized by the decrease in serum iron, serum ferritin, transferrin saturation (TSAT), and hepcidin, and the increase in total iron-binding capacity (TIBC) and soluble transferrin receptor (sTfR) in an attempt to enhance the transport of iron to tissues. At this stage, low iron leads to the formation of erythrocytes with hypochromia, which is manifested by reduced mean corpuscular hemoglobin (MCH) and reticulocyte hemoglobin content (RHC). In the stage of iron deficiency anemia, oxygen supply to tissues is damaged, which is reflected by the reduction in hemoglobin concentration, mean corpuscular volume (MCV), MCH, serum iron, serum ferritin, TSAT, RHC, and hepcidin and the increase in sTfR, TIBC, and zinc protoporphyrin.

For the diagnosis of anemia, the gold standard is hemoglobin concentrations. Hemoglobin concentration is related to age, sex, and pregnancy. Table 2 shows the cut-off values of hemoglobin concentration to diagnose anemia [4].

## 3. Iron Deficiency and Bone Loss

Iron deficiency is typically attributed to inadequate dietary intake, blood loss, or chronic inflammation and is estimated to affect more than 2 billion people worldwide, particularly women and children [31]. Due to the indispensable role of iron in numerous biochemical reactions, low body iron stores result in deleterious effects on the function of various biological systems.

### 3.1. Dietary Iron Deficiency and Bone Loss

Numerous reports from animal studies have shown that moderate to severe dietary iron deficiency alters bone microarchitecture and reduces BMD and bone strength. In 2002, Medeiros et al. [32] fed weanling male Long–Evans rats with an iron-deficient diet for 5 weeks, leading to a decrease in cortical width, cortical bone area, and BMD of the femur and tibia. In 2004, the authors found that the third lumbar trabecular bone microarchitecture in diet iron-restricted rats had a decreased bone volume fraction (BV/TV), trabecular number (Tb.N), and trabecular thickness (Tb.Th), a less favorable structural model index (SMI), and increased trabecular separation (Tb.Sp) compared with the controls [33]. Even when compared to a calcium-restricted diet, animals receiving an iron-deficient diet exhibited a lower Tb.N in the vertebrae [33]. In 2006, they further revealed the L4 vertebrae from the iron-restricted diet group had greater internal stress with an applied force than the control group using finite element analysis, indicating that low iron reduced bone strength [34]. Subsequently, Lobo et al. [35] clearly demonstrated that iron-deficient rats had lower peak load, yield load, stiffness, resilience, and absorbed energy compared to rats with iron-adequate diets. The results of these animal studies support the hypothesis that iron deficiency is associated with bone loss.

In humans, several studies have demonstrated that severe dietary iron deficiency or iron deficiency anemia has a negative effect on bone health, especially to women. In healthy postmenopausal women, a positive association between dietary iron and BMD was found [36]. However, the same authors observed in a one-year follow-up study that this association was only observed in women on hormone replacement therapy [37]. Diet potentially influences bone loss after menopause as studies have found dietary iron may have a protective effect on bone at the spine in postmenopausal women [12]. Higher bone resorption was observed in the group of young women with iron deficiency compared to the iron-sufficient group [38]. In this line, a positive association between ferritin levels and BMD was found in in elderly Korean men, but elderly women showed an insignificant positive correlation [39]. Therefore, women who suffer from iron deficiency for multiple decades may develop osteoporosis later in life.

### 3.2. Iron Deficiency Anemia and Bone Loss

Anemia resulting from iron deficiency is one of the most common and widespread health disorders in the world [4]. Low hemoglobin levels are a maker of anemia. BMD is found to be positively associated with hemoglobin levels in older men and women [40,41]. Korkmaz et al. [42] showed that anemia is associated with low BMD in postmenopausal Turkish women. However, the cause–effect relationship cannot be established due to the limitation of a cross-sectional study design in all of the studies [43]. Recently, a large population-based study indicated that patients with a history of IDA had a near two-fold risk for osteoporosis compared with individuals without anemia [10]. Similarly, a multisite longitudinal cohort study demonstrated that high BMD loss at the hip was strongly related to anemia.

Severe osteoporosis can cause fractures. A retrospective cohort study revealed that anemia is associated with increased risk of fracture in elderly men, and less strongly but still significantly in elderly women [11]. A meta-analysis also showed males with anemia had a 1.51-fold higher fracture risk, while females had a 1.09-fold higher fracture risk [44]. Similarly, Jørgensen et al. [45] demonstrated that a lower value of hemoglobin is associated with a 1.27 times higher risk of non-vertebral fracture in men and 1.08 times in women. Looker AC [46] demonstrated that both low and high hemoglobin values were associated with increased hip fracture risk in non-Hispanic whites age 65 years and older after adjusting for age and sex. Consequently, appropriate information and suggestions on osteoporosis and fracture risk should be provided to people at risk after confirmation of IDA to facilitate earlier medical management or specialty referral.

### 3.3. Iron Deficiency Anemia in Chronic Kidney Disease Complicated by Osteoporosis

IDA is a common complication of chronic kidney disease (CKD). The diagnostic criteria for anemia in patients with CKD are different from those of the general population. The KDIGO clinical practice guidelines define the diagnosis of anemia in adults with CKD and children over 15 years of age as a hemoglobin (Hb) concentration of <13.0 g/dL in men and <12.0 g/dL in women [47]. According to the National Health and Nutrition Examination Survey (NHANES) 1999–2004 populations in the USA, anemia was present in 13.9% of NHANES participants [48]. Recently, a study that included a large population of patients with CKD revealed that IDA occurs at a rate of up to 20.6% [49].

Iron deficiency associated with CKD includes both absolute and functional iron deficiency [50]. Causes of absolute iron deficiency arise from an increased iron losses, estimated at 1–3 g per year in hemodialysis patients due to frequent blood extraction and residual blood in dialysis tubes [51]. The high rate of iron loss is also because of gastrointestinal bleeding from the combination of gastritis and chronic bleeding from uremia-associated platelet dysfunction in both dialysis- and non-dialysis-dependent CKD patients [52,53,54]. In addition, CKD patients also have impaired dietary iron absorption, particularly hemodialysis patients. Indeed, oral iron is no better than a placebo in improving anemia or iron deficiency in hemodialysis patients and is less effective than intravenous iron [55]. Functional iron deficiency in CKD patients is mainly attributed to an increase in hepcidin levels caused by chronic inflammation, decreased renal clearance, and reduced EPO levels [56]. Enhanced hepcidin levels lead to a decrease in iron absorption in the intestine and a reduction in iron release from the liver and macrophages, resulting in a decrease in the available iron for erythropoiesis [57].

Osteoporosis or osteopenia is another common complication of CKD. The prevalence of osteoporosis in CKD patients ranged from 18% to 32%, while osteopenia was found in up to 57% of patients [58,59]. A significantly higher prevalence of osteoporosis and fractures has been reported in patients with end-stage renal disease (ESKD), especially in patients on hemodialysis compared to those on peritoneal dialysis [60]. Similar to the general population, women and older CKD patients are at higher risk of osteoporosis [61]. Fragility fractures are one of the most serious consequences of osteoporosis. Pre-dialysis CKD patients have a twofold increased risk of overall fracture and a fivefold increased risk of rotor fracture [62]. ESKD patients have an eightfold higher risk of fracture than the general population [63]. In addition, mortality after fracture in CKD patients is very high; for example, the mortality rate of hip fracture in dialysis patients after 1 year is as high as 64% [64].

Osteoporosis in CKD patients is multifactorial and the current view includes uremic milieu, abnormalities in vitamin D metabolism, calcium and phosphorus balance, and parathyroid hormone (PTH) as well as the use of some drugs that can cause bone loss, such as glucocorticoids, vitamin K antagonists, diuretics, and proton pump inhibitors [59,65]. It is not clear whether iron deficiency or IDA is a risk factor in the reduction of BMD in CKD patients. However, a study found that iron therapy, although effective in improving anemia in the mice with CKD, resulted in the loss of cortical and trabecular bone [66]. Therefore, the relationship between iron deficiency and osteoporosis in CKD needs to be further investigated to clarify.

## 4. The Potential Mechanism of Bone Loss Induced by Iron Deficiency

Iron is a vital component of various enzymes involved in many biological processes, including oxygen transport, DNA biosynthesis, and cellular energy generation [67]. Moreover, the biological activity of iron depends mainly on its efficient electron transfer properties. It has the ability to accept or donate electrons during the conversion between ferric iron (Fe^3+^) and ferrous iron (Fe^2+^), thereby functioning as a catalytic cofactor in various biochemical reactions [68]. Therefore, iron is necessary for the growth, proliferation, and differentiation of bone cells, especially for osteoblasts and osteoclasts. In addition, hypoxia caused by iron deficiency can also lead to the activity and function disorder of bone cells. With respect to bone metabolism, iron is involved in the synthetic process of collagen and vitamin D metabolism.

### 4.1. Physiological Iron in Bone Homeostasis

The maintenance of bone homeostasis depends on two important cells, osteoblasts that form new bone and osteoclasts that dissolve old and impaired bone. In the process of children’s growth and development, bone formation plays a major role, far greater than the speed of bone destruction, termed bone modeling in this state. Once the bone grows to maturity, these two processes will form a roughly balanced state, termed bone remodeling [69]. Appropriate regulation and coordination of the function and differentiation of osteoblasts and osteoclasts is necessary to acquire bone homeostasis. Hematopoietic stem cells, the source cells of osteoclasts, first differentiate into bone marrow mononuclear cells (BMMs) under macrophage colony stimulating factor (M-CSF) stimulation and then differentiate to form multinuclear osteoclasts in the stimulation of NF-kB ligand receptor activator (RANKL) [70]. At the initial stage of bone remodeling, mononuclear precursors are recruited to the bone surface and undergo proliferation, differentiation, and fusion to finally form mature multinucleated cells [71]. Then, osteoclasts decompose or absorb bone and release calcium into the blood. After the completion of the bone resorption process, it was widely accepted in the past that that osteoclasts would undergo apoptosis or programmed cell death [72], but recently it was recognized that osteoclasts divide into daughter cells called osteomorphs, which can fuse and re-form osteoclasts in RANKL stimulation [73]. Osteoclasts secrete a variety of signal molecules during bone resorption, attracting bone marrow mesenchymal stem cells (BMSCs) to migrate to newly excavated pits, where they proliferate and differentiate into osteoblasts [71], or activate the quiescent bone lining cells on the bone surface to form osteoblasts [74]. In the process of bone formation, osteoblasts proliferate and differentiate to maturity, and then secrete a variety of matrix proteins, such as type I collagen (COL1), alkaline phosphatase (ALP), and osteocalcin, to form osteoid [75]. Subsequently, hydroxyapatite crystals were deposited into the osteoid to form new mineralized bone. Once the mineralization process is completed, most osteoblasts are eliminated through apoptosis, and the residual are transformed into osteocytes embedded in the bone matrix and bone lining cells on the bone surface [76]. So far, the whole bone remodeling process is completed. There is no doubt that the activity and function of osteoblasts and osteoclasts are crucial in the maintenance of bone homeostasis.

Both osteoclast formation and bone absorption activity of mature osteoclasts require high energy; thus, osteoclasts contain large amounts of mitochondria. Mitochondrial respiratory complex I and peroxisome proliferator-activated receptor γ coactivator 1β (PGC-1β, key mitochondrial transcription regulator) are essential for osteoclast differentiation [77,78]. Mitochondrial reactive oxygen species (ROS) also are important components that stimulate osteoclastic differentiation and resorption of bone tissue [79]. Iron plays an important role in mitochondrial metabolism, ROS production, and the biosynthesis of heme and Fe-S clusters, which are the key components of the mitochondrial respiratory complex [80,81]. Therefore, iron is crucial for the differentiation of osteoclasts and the activation of bone resorption activity. Indeed, deferoxamine (DFO), an iron chelator, inhibits osteoclastogenesis, and bone resorption has been reported extensively in vitro [78,82,83]. Zhang et al. [82] found that DFO inhibited iron-uptake-stimulated osteoclast differentiation, negatively affected mitochondrial function through decreasing electron transport chain activity, and suppressed mitogen-activated protein kinase (MAPK) activation independently of ROS stimulation. However, the findings in vivo seem different from the results of cell experiments. In vivo, serum levels of bone turnover markers, such as tartrate-resistant acid phosphatase (TRAP), C-telopeptide of type I collagen (CTX-I), and N-telopeptide of type I collagen (NTX-I), can reflect the activity of osteoclasts [84]. Díaz-Castro et al. [85] demonstrated that severe nutritional IDA increased the levels of serum TRAP and CTX-I in rats. Toxqui et al. [38] showed that serum NTX-I levels were significantly higher in healthy iron-deficient women compared with iron-sufficient women. The reasons for discrepancies regarding the effects of iron deficiency on bone resorption between in vitro and in vivo may be related to iron-deficiency-induced hypoxia, which we will discuss in the next section.

Similarly to osteoclasts, osteoblasts also have a high demand for energy in osteogenic differentiation; thus, mitochondria play an essential role in this process [86,87]. Iron deficiency induced by treatment with an iron chelator deferoxamine (DFO) in primary cell cultures isolated from fetal rat calvaria was studied, the results of which demonstrated that low iron suppresses osteoblast phenotypic development [88]. Mild low iron induced by DFO promoted osteoblast activity, but serious low iron inhibited osteoblast activity [89]. Mouse and human progenitor cells, differentiated under standard osteoblast protocols in the presence of DFO, ALP activity, mineralization, and osteogenic genes (e.g., osterix, Col1a1, Bglap, and Dmp1), were reduced significantly [90]. Interestingly, the same dose of DFO had no significant effect on adipogenic differentiation, suggesting that osteoblasts maintain a higher iron requirement for differentiation and function compared with adipocytes [90]. In vivo, a zebrafish model of FPN gene mutant showed severe iron deficiency, resulting in a reduction in the number of calcified vertebrae in the larvae and a reduction in the expression of bone formation genes (such as alpl, runx2a, and col1a1a) and BMP signal genes (bmp2a and bmp2b), indicating that iron deficiency may inhibit bone formation through the BMP signal pathway [91]. Serum bone formation makers, procollagen type I N-terminal propeptide (P1NP) and osteocalcin, were markedly decreased by dietary iron deficiency in rats [85,92]. These parameters were recovered after supplying a normal or high-iron diet [93].

In brief, iron is involved in the regulation of the basic cell activity and function of osteoblasts and osteoclasts, and is an essential mineral element for maintaining bone homeostasis. However, iron must be controlled at the physiological level because excessive iron can lead to the over-activation of osteoclasts and the toxic effect of osteoblasts, resulting in the imbalance of bone homeostasis, and ultimately induce osteopenia and even osteoporosis [94,95].

### 4.2. Iron-Deficiency-Induced Hypoxia in Bone Homeostasis

Iron is closely associated with oxygen sensing in cells. Since iron is indispensable to oxygen transport, iron deficiency can create low-oxygen conditions (hypoxia) as a result of decreased oxygen delivery to cells and tissues. Hypoxia-inducible factors (HIFs), the heterodimeric transcription factors composed of an oxygen-sensitive α subunit and a stable β subunit, are critical mediators of the cellular response to hypoxia [96]. HIF1-α and HIF2-α are proposed to contribute to the functions of osteoblasts and osteoclasts in bone homeostasis [97,98], which can potentially be attributed to the hypoxic condition in bone microenvironments, such as the endosteal areas of bone medullary cavity and the epiphyseal growth plate [99].

Hypoxia stimulates angiogenesis-dependent bone formation during bone regeneration, mainly via VEGF-dependent promotion of neoangiogenesis through the HIF-1α pathway [100]. Mice overexpressing HIF-1α in osteoblasts increases bone modeling in developing bone [101]. Mice lacking HIF-1α in osteoblasts markedly decreases trabecular bone volume, reduces bone formation rate, and alters cortical bone architecture in developmental long bone [102]. Prolyl hydroxylases (PHDs) and Von Hippel–Lindau (VHL) are the upstream regulators of HIF-α. Under normoxia, PHDs hydroxylate HIF-α on two proline residues, inducing VHL-mediated poly-ubiquitination and proteasomal degradation [103]. Therefore, knockdown of PHDs or VHL in mesenchymal progenitors leads to elevated HIF-1α, which further stimulates osteogenesis [104,105]. However, another study showed that conditional knockout of PHD2 in osteoblast lineage cells resulted in reduced bone mineral content, bone area, and BMD in femurs and tibias of mice [106]. Bone marrow stromal cells derived from PHD2 knockout mice formed fewer mineralized nodules when cultured with a mineralized medium, suggesting that PHD2 plays an important role in regulating bone formation [106]. Several studies have shown that hypoxia and HIF-1α promote osteoclastogenesis and subsequent bone resorption during bone remodeling, in contrast to the anabolic role of HIF-1α in bone modeling. For example, hypoxia and HIF-1α overexpression inhibit BMP-2-induced osteoblast differentiation and stimulate osteoclastogenesis [107]. Osteoclast-specific HIF-1α inactivation antagonizes bone loss in ovariectomy (OVX) mice and osteoclast-specific estrogen receptor α-deficient mice, whereas oral HIF-1α inhibitors protect OVX mice from osteoclast activation and bone loss [108]. Similarly, HIF-1α protein accumulates in osteoclasts following orchidectomy (ORX) in mice, and administration of a HIF-1α inhibitor abrogated ORX-induced osteoclast activation and bone loss [109]. The reasons for potential differences regarding the role of HIF-1α in bone modeling and remodeling are unclear but may reflect the combined coordinated or coupled effects of multiple cell types in the bone microenvironment in vivo.

In contrast to the role of HIF-1α in promoting bone formation, HIF-2α inhibits the differentiation of osteoblasts [110]. HIF-2α deficiency in mice enhances bone formation in vivo and overexpression of HIF-2α inhibits osteoblast differentiation of mouse calvarial preosteoblasts by targeting *Twist2* [111]. Although both HIF-1α and HIF-2α activate osteoclasts, the former mainly mediates the bone resorption capacity of osteoclasts, while the latter is mainly involved in the process of osteoclast formation [112]. HIF-2α deficiency in mice stimulates osteoclast formation in vivo, and overexpression of HIF-2α enhances osteoclast differentiation of mouse-bone-marrow-derived macrophages (BMMs) via regulation of Traf6 [111]. In addition, HIF-2α contributes to the interaction between osteoblasts and osteoclasts by directly targeting RANKL in preosteoblasts [111]. Osteoblast-specific or osteoclast-specific conditional knockout of HIF-2α in male mice enhances bone mass and reverses age-related bone loss [111,113]. Consequently, HIF-2α may regulate bone homeostasis through its effects on osteoblasts and osteoclasts during bone remodeling.

### 4.3. Iron in Collagen Synthesis

Iron participates in a variety of enzymatic systems in the body, including the enzymes involved in collagen synthesis. In mammals, approximately 28 types of collagen have so far been identified [114]; among these types, the most prevalent is type I collagen that was found in the extracellular matrix (ECM), particularly in tissues such as tendon and bone [115]. Bone is a complex assembly of type I collagen fibers filled in with mineral crystal of hydroxyapatite [116]. Type I collagen constitutes 90% of the organic matrix, and together with the mineral, governs the biomechanical properties and functional integrity of bone [117].

For collagen synthesis, a three-dimensional stranded structure with the amino acids glycine and proline as its principal components is assembled primarily. This is called procollagen and is precursor of collagen. Procollagen is then modified by adding the hydroxyl groups to proline and lysine. This reaction requires α-ketoglutarate, molecular oxygen, ferrous iron, and a reducing agent, and it is an important step to later glycosylation and the formation of the triple-helix structure of collagen [118]. In this process, ascorbate and molecular oxygen reduce the inactive Fe^3+^ to the active Fe^2+^ [119], and α-ketoglutarate is decarboxylated oxidatively to produce succinate and CO_2_ [120]. Two different enzymes, prolyl-hydroxylase and lysyl-hydroxylase, catalyze these hydroxylation reactions, and iron is essential in this pathway. Moreover, iron-containing prolyl hydroxylases are also involved in the regulation of hypoxia-inducible factors (HIF), which sense oxygen status in the body. When oxygen is sufficient, HIFs are transcriptionally restrained through ubiquitination by prolyl hydroxylation [121]. Therefore, it is a hypothesis that in an iron deficiency status, there may be less iron available to the prolyl and lysyl hydroxylases which could result in decreased cross-linking activity and, subsequently, weaker collagen fibers [30]. 

C-terminal propeptide of type I procollagen (P1CP) and P1NP, the byproducts of the process of type I collagen synthesis, are cleaved from type I procollagen by proteases outside the osteoblast [84]. Serum P1NP and P1CP levels reflect the rate of type I collagen synthesis and osteoblast activity. CTX-I and NTX-I are the degradation products of collagen, and serum CTX-I and NTX-I levels reflect the rate of type I collagen degradation. Iron deficiency can decrease serum P1NP or P1CP levels and increase serum CTX-I or NTX-I levels, which has been shown in human and animal studies. Wright et al. [122] indicated that serum NTX-I levels were significantly higher and serum P1NP levels tended to be lower in woman with iron deficiency anemia (IDA). Similarly, the lower amount of serum P1NP and increased serum CTX-I were found in IDA rats by Díaz-Castro et al. [85]. Scott et al. [123] also demonstrated that iron deficiency increased the concentration of serum CTX-I in rats. Therefore, iron deficiency may reduce the synthesis and increase the degradation of type I collagen in bone tissue.

### 4.4. Iron in Vitamin D Metabolism

Iron is involved in bone metabolism, and another mechanism may be through vitamin D activation and deactivation [124,125]. Vitamin D plays a major role in the regulation of mineral homeostasis and affects bone metabolism [126]. The vitamin D hormone maintains a constant level in serum calcium concentrations. Active 1,25-dihydroxyvitamin D [1,25(OH)_2_D] promotes the absorption of calcium and phosphorous in the intestine, phosphate reabsorption in the kidney, and calcium and phosphate release from the bone [127]. Moreover, active vitamin D has a direct regulating effect on the activity and function of osteoblast and osteoclast [128,129,130]. Therefore, vitamin D deficiency in the body will cause osteoporosis; in contrast, vitamin D supplementation can increase bone density and is widely used in the treatment of osteoporosis [131,132].

Iron is an essential element involved in the cytochrome P450 superfamily, which catalyzes single or multiple hydroxylation reactions in pathways by vitamin D substrate at specific carbons using a heme-bound iron [125]. Vitamin D from the diet and skin has two steps of hydroxylation for its activation. Firstly, it is hydroxylated in the liver into 25-hydroxyvitamin D [25(OH)D] by the cytochrome P-450 25-hydroxylase (CYP2R1) [133]. In addition, further hydroxylases, such as CYP27A1 and CYP3A4, have been found involved in the anabolism of vitamin D in the liver [134,135]. In the second step, 25(OH)D is transported to the kidney and is hydroxylated into 1,25(OH)_2_D, the active form of the vitamin D, by the 25-hydroxyvitamin D 1α-hydroxylase (CYP27B1). In the inactivation process, the 1,25(OH)_2_D is inactivated by the 1α,25-hydroxyvitamin D 24-hydroxylase (CYP24A1) through multiple oxidations of the sterol side chain [136]. In case of iron deficiency in tissues, the activity of iron-containing enzymes is decreased [137].

Numerous studies have revealed a relationship between clinical iron deficiency and low vitamin D levels [38,138]. Blanco-Rojo et al. [138] demonstrated that vitamin D deficiency or insufficiency is very high in women with iron deficiency. However, the recovery of iron status by an iron-fortified diet did not affect 25(OH)D levels. Grindulis et al. [139] showed that there is a significant association between iron deficiency and lower vitamin D levels in Asian children. Qader et al. [140] revealed that serum vitamin D levels were lower in Iraqi children with iron deficiency in contrast with children with normal iron. El-Adawy et al. [141] certified that vitamin D deficiency had a higher frequency in Egyptian adolescent females with IDA than the healthy control. Jin et al. [142] found that vitamin D deficiency existed in 67% of infants with IDA. Thus, vitamin D supplementation is important particularly in IDA infants. Ferritin, an iron-containing protein with high molecular weight, plays a key role in the body as an iron storage compound [143]. Serum ferritin concentration has been proposed as an index of iron deficiency and iron overload [144]. Kang et al. [145] showed that there is an association between lower serum ferritin and vitamin D in breastfed infants and their mothers that had anemia during pregnancy.

In animal models, an iron-deficient diet has been shown to reduce serum vitamin D levels. Rats were fed on an iron-deficient diet for 6 weeks, and the serum levels of 25-(OH)D and 1,25-(OH)_2_D were significantly lower than those in rats with a normal diet [146]. Moreover, Western blotting, immunofluorescence, and q-PCR assay revealed that the protein and gene expressions of CYP2R1, CYP27A1, and CYP24A1 in the iron-deficient diet group were down-regulated compared to control group, and the expression of CYP27B1 protein and gene was up-regulated in rats with a low-iron diet. However, another study indicated that an iron-deficient diet reduced renal CYP27B1 activity, accompanied by a decreased serum 1,25-(OH)_2_D_3_ concentration and bone formation in rats [147]. These data demonstrate that iron may be involved in the metabolism of vitamin D by regulating the expression of vitamin D hydroxylase, suggesting that appropriate iron supplementation might activate vitamin D.

In addition, iron deficiency regulates vitamin D metabolism potentially through fibroblast growth factor 23 (FGF23), which is secreted by osteocytes [148]. The main target organ of FGF23 is the kidney, where FGF23 inhibits transcription of the key enzyme in vitamin D hormone activation, CYP27B1, and activates transcription of the key enzyme responsible for vitamin D degradation, CYP24A1 [149]. In the absence of FGF23 signaling, tight control of renal 1α-hydroxylase fails, leading to overproduction of 1,25-(OH)_2_D in mice and humans [150,151]. Therefore, FGF-23 is a crucial modulator of vitamin D metabolism. Iron deficiency stimulated the transcription of FGF23 in osteocytes and the increase in serum levels of intact, biologically active FGF23 (iFGF23) and C-terminal fragments, a degradation product; biologically inactive FGF23 (cFGF23) in mice has been reported [152,153]. However, several studies revealed that serum iFGF23 levels did not change, whereas cFGF23 levels were significantly increased in iron-deficient mice [152,154]. Similar alteration of serum FGF23 levels was also found in patients with iron deficiency or IDA [155,156]. FGF23 is metabolized and eliminated by the kidneys [157]. Therefore, we speculate that the increased FGF23 is degraded by the kidneys after regulating vitamin D metabolism in the kidneys. Indeed, mice with iron-deficient diets showed an increase in bone FGF23 mRNA accompanied by an increase in Cyp24a1 mRNA and a decrease in Cyp27b1 mRNA in the kidneys, resulting in a decrease in serum 1,25(OH)_2_D [154].

In conclusion, iron deficiency can directly or indirectly regulate the activity and function of osteoblasts and osteoclasts by inducing hypoxia and disorder of vitamin D metabolism, ultimately destructing bone homeostasis (Figure 2). Reduced active vitamin D also leads to disturbances in calcium and phosphorus metabolism and thus affects bone metabolism. In addition, iron deficiency potentially inhibits new bone formation by deranging collagen synthesis.

## 5. Prevention of Iron Deficiency

There are four main strategies to correct iron deficiency in the population, including dietary modification to improve iron intake and bioavailability, iron supplementation (oral or intravenous), iron fortification of foods, and new approaches to biofortification, which can be used individually or in combination to correct iron deficiency. However, when considering iron, there are some difficulties in applying these strategies.

Dietary modifications include increasing the intake of those foods that promote iron uptake, especially fresh ascorbic-acid-rich fruits and vegetables, to enhance the absorption of non-heme iron, and decreasing the intake of those foods that inhibit the absorption of non-heme iron, such as tea and coffee [158]. The phytic acid in cereal and legume supplements inhibits the absorption of iron [159]. Therefore, increasing the activity of endogenous or exogenous phytase through techniques such as germination and fermentation or using non-enzymatic methods such as thermal processing, soaking, and milling can reduce the content of phytic acid in cereals and legumes to improve the bioavailability of iron [160,161]. 

For oral iron supplementation, ferrous salts, such as ferrous sulfate and ferrous gluconate, are preferred because of their low cost and high bioavailability [162]. Since food reduces the absorption of medicinal iron by about two-thirds, it should be taken on an empty stomach whenever possible [163]. Patients who are intolerant or unresponsive to oral compounds may opt for intravenous (IV) iron. The advantage of IV iron is that it works quickly and has negligible gastrointestinal toxicity [164]. For example, for the treatment of anemia in patients with CKD, IV iron is more effective than oral iron and avoids oxidative damage to the intestinal mucosa in patients with active inflammatory bowel disease [165,166]. Notably, iron oxide nanoparticles (IONs), such as ferumoxytol, approved for iron supplementation in CKD anemia patients will bring additional benefits to anemia patients [167]. For example, ferumoxytol has been shown to be used in the treatment of other diseases, including leukemia and osteoporosis [168,169,170]. Therefore, IONs are expected to be the most promising drugs for the treatment of iron deficiency complicated by osteoporosis.

The most bioavailable are those iron compounds that are soluble in water or dilute acids, but they usually react with other food components, resulting in poor taste, color changes, or fat oxidation [171]. Therefore, less soluble iron is usually chosen for fortification to avoid unwanted organoleptic changes, although these are not well absorbed [172]. Fortification is usually made with a much lower dose of iron than supplementation, closer to the physiological environment. The iron compounds recommended by the WHO for food fortification include ferrous sulfate, ferric pyrophosphate, ferrous fumarate, and electrolytic iron powder [173]. Wheat flour is the most common iron-fortified food. Some commercial infant foods, such as formula and cereals, often contain iron as well.

Biofortification strategies include plant breeding and genetic engineering [174]. For example, the iron content in rice endosperm was enhanced by Lucca et al. through genetic engineering to improve its absorption in the human intestine [175]. They introduced a ferritin gene from kidney beans into rice to triple its iron content, while they introduced a heat-tolerant phytase from Aspergillus fumigatus into rice endosperm to improve iron bioavailability. Masuda et al. [176] developed an iron-biofortified rice by using three transgenic approaches. These transgenic seeds grown in greenhouses produced rice with six times higher iron concentration than non-transgenic seeds and 4.4 times higher than non-transgenic seeds grown in a paddy field.

## 6. Conclusions

Based on current knowledge, we summarized the negative effects of iron deficiency with and without anemia on bone and analyzed the possible mechanisms. However, there are many unanswered questions regarding the hypothesis that chronic iron deficiency predisposes patients to reduced bone mass, osteoporosis, and fracture risk. It is not clear to what extent severe or mild iron deficiency affects the bones. The mechanism of iron-deficiency-induced bone loss should be more thoroughly explored. For instance, the protective or aggravating effects of different hormones (EPO, hepcidin, etc.), inflammation, acidosis, and anemia or other causes of hypoxia should be investigated. In addition to cellular and animal studies, epidemiological investigations, and retrospective study analysis, the main work should be performed in human prospective studies. Hematological parameters and bone quality assessment should be measured over several years in different populations such as children and adolescents, women of childbearing age and the elderly with reliable endpoints to understand more clearly the relationship between iron deficiency and osteoporosis. Importantly, ferumoxytol deserves to be experimented or trialed on animals or patients with IDA other than CKD to expand their broader application for anti-anemia and anti-osteoporosis.

## Figures and Tables

**Figure 1 ijms-24-06891-f001:**
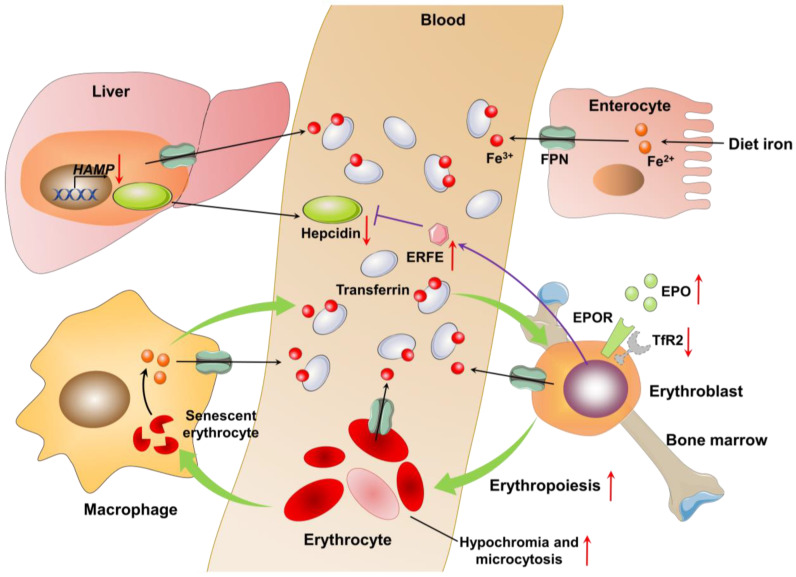
Iron homeostasis in iron deficiency. Most iron in the body is involved in the process of erythropoiesis, phagocytosis, and dissolution of senescent erythrocytes by macrophages (light green arrows). Hepcidin, produced and secreted by hepatocytes, is the master regulator of systemic iron metabolism. In iron deficiency, the transcription of the hepcidin-encoding gene, HAMP, is down-regulated. Low hepcidin levels increase recycling by macrophages and iron absorption by enterocytes by increasing the activity of iron exporter ferroportin (FPN). Low hepcidin levels also lead to the efflux of iron from erythroblasts via FPN, which further reduces the intracellular iron availability of erythroblasts. Iron deficiency stimulates erythropoiesis by enhancing the production of erythropoietin (EPO) and erythroblastic EPO sensitivity via the genetic loss of the EPO receptor (EPOR) partner transferrin receptor 2 (TfR2). The increase in erythropoiesis and the decrease in iron supply in erythroblasts lead to an increase in erythrocytes with hypochromia or microcytosis. Erythroferrone (ERFE), a hormone that EPO stimulates erythroblasts to produce, can also inhibit the production of hepcidin. The red up arrows mean elevation; The red down arrows mean lowering; Black arrows represent hepcidin-FPN axis-regulated iron homeostasis; Purple arrows represent erythroblast-regulated hepcidin expression; Green arrows represent the iron cycle during erythropoiesis.

**Figure 2 ijms-24-06891-f002:**
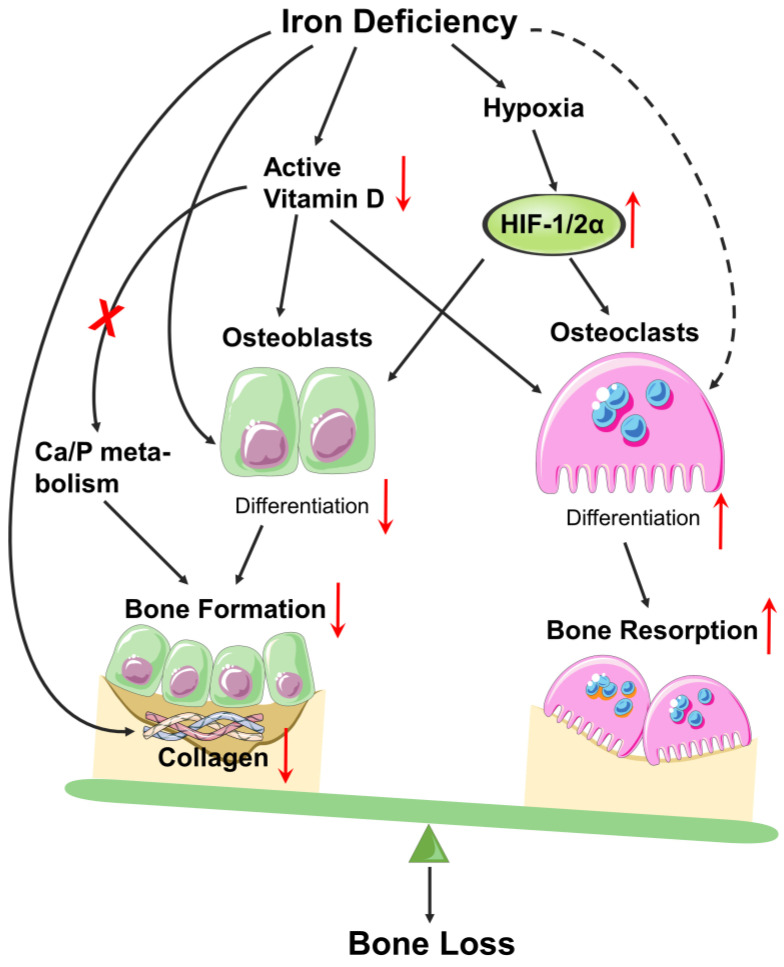
Schematic diagram of potential mechanisms of iron-deficiency-induced bone loss. Iron deficiency directly or indirectly inhibits osteoblastic differentiation and stimulates osteoclastic differentiation by resulting in hypoxia-induced elevated HIF-1/2α expression and decreasing active vitamin D levels. Decreased osteoblast differentiation and activated osteoclast differentiation leads to decreased bone formation and increased bone resorption, respectively. Reduced active vitamin D also leads to disturbances in calcium (Ca) and phosphorus (P) metabolism, thereby reducing bone formation. In addition, iron deficiency may directly inhibit new bone formation by decreasing collagen synthesis. The red up arrows mean elevation; The red down arrows mean lowering; The X-type represents disruption.

**Table 1 ijms-24-06891-t001:** Biochemical markers for identifying iron deficiency.

Markers	Normal	Iron Depletion	Iron Deficiency without Anemia	Iron Deficiency Anemia
**Current**				
Hemoglobin (g/dL)	N	N	N	L
Mean corpuscular volume (fL)	N	N	N	L
Mean corpuscular hemoglobin (pg)	N	N	N/L	L
Ferritin (μg/L)	N	L	L	L
Serum iron (μg/dL)	N	N	L	L
Transferrin saturation (%)	N	N	L	L
**Proposed**				
Reticulocyte hemoglobin content (pg)	N	N	L	L
Soluble transferrin receptor (mg/L)	N	N	H	H
Total iron-binding capacity	N	N	H	H
Hepcidin	N	N/L	L	L
Zinc protoporphyrin	N	N	N	H

N—normal; H—high; L—low.

**Table 2 ijms-24-06891-t002:** Hemoglobin levels (g/L) to diagnose anemia at sea level.

Population, Age	No Anemia	Anemia		
		Mild	Moderate	Severe
Children, 6–59 months	≥110	100–109	70–99	<70
Children, 5–11 years	≥115	110–114	80–109	<80
Children, 12–14 years	≥120	110–119	80–109	<80
Non-pregnant women, 15 years and above	≥120	110–119	80–109	<80
Pregnant women	≥110	100–109	70–99	<70
Men, 15 years and above	≥130	110–129	80–109	<80

## Data Availability

Not applicable.

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
