# Peer review of "Iron Deficiency and Iron Deficiency Anemia: Potential Risk Factors in Bone Loss"

_ijms, 2023, doi:10.3390/ijms24086891_

Round 1

Reviewer 1 Report

IJMS – Iron deficiency and iron deficiency anemia: potential risk factors in bone loss

Review article about the intersection of iron deficiency and potential pathogenic effects on bone.

Abstract:

I have no comments for this section.

Introduction:

Page 2, line 70-73: There is possibly an extra “of hemoglobin” within the second sentence. It would flow better if you edited this specific sentence such that “much of the remaining iron is stored….” was the start of a new sentence.

Page 2, line 80-82: Please edit the sentence “Because there is no an excretion regulation pathway…..”

Page 5, line 164: Please edit the wording of the beginning part of the sentence “With respect regard human,”

Page 5, line 169-170: The wording of this sentence is a bit awkward my suggestion: Diet potentially influences bone loss after menopause as studies have found dietary iron may have a protective effect of bone at the spine in post-menopausal women.

Page 8-9: As you are discussing the role of HIF1α in bone, I think it would be prudent to describe studies that demonstrate deletion of upstream regulators (PHDs and VHL) in bone, enhance bone formation.

Figures:

Figure 1 caption, line 111-113: Please edit the sentence as the wording is a bit awkward “low hepcidin levels also lead to erythroblasts and erythrocytes donate iron….”   

Author Response

Dear reviewer,

we appreciate you very much for your positive and constructive comments on our manuscript. For your kind comments, we respond step by step as follows, and the changes in the manuscript are indicated by words in red.

Point 1: Page 2, line 70-73: There is possibly an extra “of hemoglobin” within the second sentence. It would flow better if you edited this specific sentence such that “much of the remaining iron is stored….” was the start of a new sentence.

Response 1: Yes, there was a duplication of "of hemoglobin" and we have deleted it. Based on your suggestion, we have modified the sentence " much of the remaining iron is stored…." into a separate sentence, as follows " Most of the remaining iron is stored in the liver as ferritin, and the rest is found in myoglobin and enzymes".

Point 2: Page 2, line 80-82: Please edit the sentence “Because there is no an excretion regulation pathway…..”

Response 2: We have edited the sentence “Because there is no an excretion regulation pathway…..” to “Since the body lacks a regulatory pathway for iron excretion, a balanced regulation of dietary iron intake, intestinal iron absorption and iron cycling is necessary.”

Point 3: Page 5, line 164: Please edit the wording of the beginning part of the sentence “With respect regard human,”

Response 3: We have edited the sentence “With respect regard human,” to “In humans”.

Point 4: Page 5, line 169-170: The wording of this sentence is a bit awkward my suggestion: Diet potentially influences bone loss after menopause as studies have found dietary iron may have a protective effect of bone at the spine in post-menopausal women.

Response 4: A good suggestion, we have replaced it with the sentence you proposed

Point 5: Page 8-9: As you are discussing the role of HIF1α in bone, I think it would be prudent to describe studies that demonstrate deletion of upstream regulators (PHDs and VHL) in bone, enhance bone formation.

Response 5: As per your suggestion, we have described it in the second paragraph of Subsection 4.2.

Point 6: Figure 1 caption, line 111-113: Please edit the sentence as the wording is a bit awkward “low hepcidin levels also lead to erythroblasts and erythrocytes donate iron….”  

Response 5: We have edited the sentence “low hepcidin levels also lead to erythroblasts and erythrocytes donate iron….” to “Low hepcidin levels also lead to the efflux of iron from erythroblasts via FPN, which further reduces the intracellular iron availability of erythroblasts.”

Reviewer 2 Report

The authors summarized from literature the influence of iron on bone physiology and pathology. The review is written with establishe knowledge and in good english. I don't have any remarks.

Author Response

Dear reviewer,

we appreciate you very much for your review of our manuscript.

Reviewer 3 Report

The authors give an in-depth review of the body's iron metabolism and a more detailed description of iron's role in the bone's structure. The study is easy to read and understand, and its length is acceptable. The number and type of references are proper, and many up-to-date papers are listed. The problem they deal with is severe worldwide, and it is good that there are suggestions to solve or at least help with it. This work is about a rarely mentioned area though many people are affected by osteoporosis, even in developed countries.

My questions: it is mentioned in the paper that the liver synthesizes the majority of hepcidin hormone. There are other tissues where hepcidin is also produced, though in smaller amounts, and often, it acts locally. Is it known whether any cells of the bone structure can synthesize hepcidin?

The iron metabolism and vitamin D level in people is correlated. Is it possible that occasionally in those cases where the iron level is low in the blood, vitamin D supplementation is inadequate also?

After answering my questions, I accept the paper for publication.

Author Response

Dear reviewer,

we appreciate you very much for your review of our manuscript. For your questions, we respond step by step as follows,

Point 1: it is mentioned in the paper that the liver synthesizes the majority of hepcidin hormone. There are other tissues where hepcidin is also produced, though in smaller amounts, and often, it acts locally. Is it known whether any cells of the bone structure can synthesize hepcidin?

Response 1: Currently, hepcidin has been found to be expressed in bone marrow cells (Peyssonnaux C et al. TLR4-dependent hepcidin expression by myeloid cells in response to bacterial pathogens. Blood. 2006,107(9):3727-32. doi: 10.1182/blood-2005-06-2259.). To our knowledge, there are no relevant studies on hepcidin expression in any bone cells.

Point 2: The iron metabolism and vitamin D level in people is correlated. Is it possible that occasionally in those cases where the iron level is low in the blood, vitamin D supplementation is inadequate also?

Response 2: Yes, such cases are possible. But the low vitamin D levels in iron-deficient patients are the result of epidemiological investigations, and occasional cases of inadequate vitamin D supplementation do not affect the correlation between iron deficiency and vitamin D deficiency.